# Is the Focal Muscle Vibration an Effective Motor Conditioning Intervention? A Systematic Review

**DOI:** 10.3390/jfmk6020039

**Published:** 2021-04-28

**Authors:** Luigi Fattorini, Angelo Rodio, Vito E. Pettorossi, Guido M. Filippi

**Affiliations:** 1Department of Physiology and Pharmacology “V. Erspamer”, Sapienza University of Roma, Piazz.le A. Moro, 5, 00185 Roma, Italy; 2Department of Human Sciences, Society and Health, University of Cassino e Lazio Meridionale Via S. Angelo—Località Folcara, 03043 Cassino, Italy; a.rodio@unicas.it; 3Department of Medicine and Surgery, University of Perugia, Piazza dell’Università, 1, 06123 Perugia, Italy; vito.pettorossi@unipg.it; 4Department of Neuroscience, Università Cattolica del Sacro Cuore, 00168 Rome, Italy; guidomaria.filippi@unicatt.it

**Keywords:** power, force, efficiency, motor adjustments, muscle vibration

## Abstract

Mechanical vibration, applied to single or few muscles, can be a selective stimulus for muscle spindles, able to modify neuromuscular management, inducing short and long-term effects, are now mainly employed in clinic studies. Several studies reported as treatments with focal vibratory (FVT) can influence neuromuscular parameters also in healthy people. However, the application modalities and the consequent effects are remarkably fragmented. This paper aims to review these studies and to characterize the FVT effectiveness on long-term conditional capacities in relation to FVT characteristics. A systematic search of studies published from 1985 to 2020 in English on healthcare databases was performed. Articles had to meet the following criteria: (1) treatment based on a locally applied vibration on muscle belly or tendon; (2) healthy adults involved; (3) outcomes time analysis enduring for more than 24 h. Twelve studies were found, all of them presented an excellent quality score of ≥75%. All selected papers reported positive changes, comparable with traditional long-lasting training effects. Muscle force and power were the most investigated parameters. The after-effects persisted for up to several months. Among the different FV administration modalities, the most effective seems to show a stimulus frequency of ≈100 Hz, repeated more times within three-five days on a voluntary contracted muscle.

## 1. Introduction

The production and control of a human voluntary movement is a complex process involving both nervous and muscular systems. Muscular performance changes are obtained by acting a tune of activation between these two systems. In this regard, a well-defined and structured intervention able to change one or more performance parameters for a long time is often called motor conditioning. Traditionally, this is achieved by means of training programs composed by an adequate exercise stint and workload [1]. These programs require remarkable compliance from the subject and long-lasting training application for enhancing force, motor endurance or ability [1,2]. Moreover, intervention based on exercise execution could not fit with a subject’s eventual status, because of a muscular frailty or exercise intolerance (e.g., elderly, post trauma subjects). To overcome these limits, in the main attempt both to reduce the training duration and to enhance the subjects’ compliance and performance, in recent years, different interventions based on electrical [3] or mechanical vibratory stimulation have been proposed.

Regarding the mechanical stimulus, the most adopted technique, particularly in sport sciences, consists of an elicitation, typically starting from the feet to the whole body (WBV) [4]. The rational of the WBV approach is based on the ability of the mechanical vibratory stimulus to excite muscular proprioceptors, particularly muscle spindles [5,6,7,8,9]. These latter exert a decisive influence on the motor control and could induce a subsequent neural conditioning [10,11]. Even if a large consensus exists on the effectiveness of WBV to induce positive motor effects [12,13,14], some authors have reported no effects [15], or potential harmful consequences [16]. Moreover, WBV does not allow action on single muscles, or on a muscular group, a condition that is often desirable in athletes’ training or in muscle reconditioning post trauma.

The focal vibration (FV) is another modality to apply mechanical vibration, and it allows us to locally act on well-defined body part. Many studies have been carried out to assess the FV effects on motor parameters both in healthy and pathologic individuals. In this respect, a large variety of FV treatments (FVT), different from each other in the administration modalities and the tested functional parameters, are present in the literature. As an example, FVT durations differ from a single day to several weeks, and the application number varies between one and more times per week. The muscle state may be contracted or relaxed during stimulation. The tested performance parameters are different, belonging to several physiological functionalities, from the simple force amplitude to the force time course or movement coordination, with outcome follow-ups ranging from hours to months [17,18,19,20,21,22,23,24,25]. Such protocol differences are likely the origin of different reported results. Even if positive effects seem to stand out, they vary according to the after-effect type, magnitude, persistence. The differences among the tested protocols demand definition of the possible beneficial applicative effects and the most effective stimulus characteristics. For this reason, by analyzing the literature on FVTs, some authors have tried to clarify the possible role of FVT in motor disease therapy [26]. Whereas others have been provided to illustrate the acute and chronic neuromuscular changes as the effect of FV [27]; however, these authors concluded that it is still difficult to define practical recommendations.

Therefore, the present paper aims to review the literature in order to grow knowledge on the efficacy of FVT for inducing persistent (at least 24 h from treatment ending) enhancements of conditional (those regarding metabolic processes are strictly dependent on the anatomical and physiological characteristics of some apparatuses, and are distinguished in: strength, endurance, speed and flexibility, understood as joint mobility and muscle stiffness) abilities in healthy subjects. The choice to limit the analysis of healthy subjects is to avoid the presence of any pathology as a possible confounding factor in evaluating the physiological effects of FV. Attention is also focused on looking for a dose-effect relationship between the motor parameters’ changes and the intervention characteristics (i.e., stimulus frequency, administration modalities and muscle state during treatment). Finally, the interpretative suggestions forwarded by the different studies are compared to hypothesize a possible common mainframe for considering the FVT as motor conditioning, with applications in motor and sport sciences.

## 2. Materials and Methods

The Preferred Reporting Items for Systematic reviews and Meta-Analyses (PRISMA) guidelines were followed in this review [28].

### 2.1. Data Sources

A systematic literature search was conducted, from January 1985 to June 2020, in the online databases PubMed, Web of Science, and The Cochrane Library. The following Medical Subject Headings (MeSH) of the United States National Library of Medicine (NLM) and search terms were included in our Boolean search syntax: (focal OR local OR segmental) AND vibration AND (healthy OR player OR athlete). The search was limited to English language, human species, full text availability and only original studies. Chain searching by backtracking relevant publications, scanning reference lists of relevant articles, on the basis of the authors’ knowledge and consultations with experts, in the field were performed systematically.

### 2.2. Selection Criteria

Two reviewers (LF, GMF) extracted relevant data according to a structured script from each study. The structured script included abstract, study design, sample characteristics (sample size, sex), experimental and control group characteristics, outcome measure and timing of results. Inclusion criteria were decided by the consensus statements of these two reviewers. In cases where LF and GMF did not reach agreement on inclusion of an article, AR and VEP were contacted. Inclusion criteria were selected by (a) population: healthy subjects, that is, subjects recruited did not have any illnesses or impairment that prevented the possibility of carrying out a physical activity aimed at improving conditional capacity; (b) intervention: treatment adopting localized mechanical vibration with a clear a description of apparatus; (c) outcomes: neuromuscular parameters regarding conditional abilities; (d) effects duration: at least 24 h from treatment ending.

### 2.3. Study Eligibility

Studies were excluded if they (a) analyzed outcomes in a short follow up period (<24h); (b) the outcomes were not regarding conditional motor abilities; (c) applied FVT by summing up the FV application with some type of physical exercise, in the same applicative session or in the period from treatment ending to the subsequent 24 h; (d) the administration of the stimulus involves, at the same time, many different muscles; (e) did not present an original investigation (reviews or proceedings); (f) did not publish in the English language.

### 2.4. Assessment of Methodological Quality

The study quality of each publication was evaluated by LF, GMF, AR, VEP, according to the 16-item checklist Downs and Black modified scale [29]. The quality scores were classified as “low” methodological quality for scores ≤50%; “good” for scores between 51% and 75%; and “excellent” if the score was >75%.

## 3. Results

Following the Boolean search syntax (see above) a total of 748 study records were identified. From these, 654 were excluded through title-abstract assessment and 34 were discarded as duplicates. From the remaining 60 studies without duplications, 50 ones were discarded verifying the study eligibility. Two studies were added, resulting from the chain searching. Consequently, a total of 12 eligible studies were included in the present review [30,31,32,33,34,35,36,37,38,39,40,41]. The study selection flow chart is shown in Figure 1.

### 3.1. Assessment of Methodological Quality

According to the assessment of methodological quality, three studies showed a score of 75%, seven ranged between 81% and 87%, two studies resulted in scores of over 90%. The items showing the lowest score concerned with the evaluation of the statistical power size (two out 12 studies) and the number of dropouts (stated in three out of 12 papers). The inter-rater reliability analysis showed a good coherence among the observers, 0.91 being the kappa value.

### 3.2. Outcome Measures and After-Effects

The main outcomes are the maximal voluntary contraction (MVC) and the muscle power. However, some studies analyzed the force output decline (fatigue) and the body balance. All the studies reported statistically significant improvements of, at least, one tested parameter. On the other hand, no side effects were evidenced during and after FVT in the different studies. Follow-up ranged between 24 h and 360 days, without treatment repetitions. The twelve selected studies and their main points, analyzed in the present review, are summarized.

### 3.3. Bias

In relation to the review’s aims, some biases were recognized in the selected papers. The different studies, commonly, paid poor attention to describe all the modalities of FV administration. In this respect, the features relative the contact between the stimulatory apparatus and the tissues are often absent. Since such detail might deeply influence the transmission of the mechanical stimulus to the muscle receptors, it constitutes a variable which might alter the definition of the dose-effect relation. Moreover, in some papers, the follow-up was not large enough to define the after-effect persistence. In this regard, some studies do not clearly describe the level of physical activity before and after FVT and, consequently, it cannot be excluded that the effect duration was due to a possible increase of physical activities exerted after the treatment.

## 4. Discussion

The aim of this review is to verify whether an intervention adopting FV could be considered a motor conditioning, that is, inducing long-lasting motor performance changes. Moreover, the review attempts to identify the relationship between protocol and parameter changes, by analyzing the intervention modalities. Twelve papers were listed, after a literature selection including only trials enrolling healthy adults (see above, selection criteria), with outcomes measured at least 24 h after intervention ending, the mechanical focal stimuli being the only adopted treatment. In all of these, stimulus characteristics and administration protocols were well defined. It seems primarily and immediately relevant that all trials reported positive and persistent neuromuscular changes and none of the tested individuals reported negative collateral effects. Moreover, only in two cases were improvements obtained by stimulating three or five muscles of the same limb [35,38], whereas in the other studies, improvements were evoked by treating only one muscle.

### 4.1. Outcomes

As expected, a quick analysis revealed that the maximal muscular strength and power were the more investigated motor parameters and the quadriceps muscle was the more extensively tested muscle. In this regard, MVC increased up to ≈+30% in young subjects [35], whereas, in the elderly, MVC improvements of ≈+40% and ≈+50% were reported, respectively, in males and females [33]. It is noticeable that another study, not included since the evaluation was performed only immediately after the end of FVT [42], supported the MVC increase reported in a previous study by the same authors [23]. Only in a single study did MVC not increase, even if the rate of the force development (RFD) improved (≈+27%) [30]. Muscle power also evidenced significant and persistent after-effects. In volleyball players, after FVT on both *quadriceps* muscles, muscle power was evaluated by the squat jump displacement (SJ), a typical Sport Sciences functional test. In this study, at the end of the follow-up period (240 days), muscle power improved by ≈+26%. On the other hand, in counter movement jump (CMJ), power was enhanced by ≈+13%. These values in treated subjects were significantly different from those of the untreated participants, even if both followed the same team-training protocol; the control group, at the end of the follow-up, respectively improved by ≈+9% (SJ) and by ≈+7.5% (CMJ) [36]. FVT effects were also tested in a high intensity effort as in the Wingate test (WnT), repeated more times with short rest periods, able to simulate a Sprint Interval Training [41]. These authors reported a 10% increase in the power peak and of 8% in total exercise work performed. The Jump task also showed in the elderly (≥65 years old), at the end of the follow-up (90 days), an improvement of about +40% [32,36].

### 4.2. Focal Vibration and Fatigue

Aside from maximal muscular strength and power, fatigue was also investigated after FVT. The decline of the motor output, induced by prolonged muscle activity, an important parameter in the sportive performance, as well as a relevant limiting factor in rehabilitative practice, was investigated in two studies, before and after FVT, by analyzing both the mechanical performance [30,38] and the myoelectric manifestations of fatigue [31]. The mechanical performance, evaluated by the number of the motor task repetitions, improved by about +40% [30] and +30% [38], whereas the fatigue index, explored by an analysis of the myoelectrical signals, improved by about +20% [31]. It is remarkable how, the protocols being different, the first two studies respectively evidenced such improvements 15 and 10 days after the end of FVT, whereas, in the third study, the follow-up was after 48 h.

### 4.3. After-Effect Duration

Some authors tested at different intervals the FVT results, allowing a gross time course of the after-effects [30,32,35,37,38,40,41]. The FVT after-effects, with respect to their temporal development, commonly show an early significant manifestation, as stated by the authors, after a few minutes [35,38,41], or after 24 h [32,36,37], even if two studies, with FVT different protocols with respect to the others, evidenced by the earliest statistically significant results after only two or four weeks of FVT [33,39]. Two studies extended the follow-up and a surprisingly long persistence of the FVT after-effects was evidenced by Brunetti’s group, up to 240 [36] and 360 days [37]. These data seem to be confirmed by other authors too, showing the absence of any significant decay of the after-effect throughout the follow-up period, ranging in these reports between 15 days and 16 weeks [32,33,35,39,40,41]. These long-lasting after-effects might be reasonably sustained by some other factors related to an improved motor efficiency, due to FVT, eliciting execution of a spontaneous higher workload daily. This hypothesis could be particularly reasonable in athletes [36] and elderly people [37], even if participants were asked not to change their lifestyle. In particular, in elderly people it is reasonable to suppose the onset of a virtuous circle induced by better self-confidence related to balance improvements after FVT [43].

### 4.4. FVT Versus Traditional Training Protocols

Evidence, concerning almost all papers, shows that the magnitudes of the FVT after-effects are roughly comparable with the improvement levels commonly obtained by the traditional long-lasting training protocols. As an example, eight weeks of training of leg extensions are reported as necessary to induce +27% in MVC [1]. Also, it is worth noting the performance improvements obtained in elderly people on force parameters [32,33,37], since, in this population, power and strength ameliorations can only be improved after traditional training protocols, requiring high compliance and long duration [44,45]. In the same elderly population, several papers showed an effectiveness of FVT on balance parameters [32,37], by treating quadriceps muscles and this result appears to be potentially important, the body equilibrium being in decline in elderly people related to fall risk [46]. In this regard, other studies, here excluded because of the inclusion criteria, supported, with large samples, FVT as a valid strategy to counteract falls in the elderly population [43,47].

### 4.5. Protocol Parameters

Since the different studies varied the FV application modalities, to better focus on FVT applicability, some considerations must be forwarded to better define the FV parameters which can influence the intervention efficacy. Attention was focused on the vibration frequency, on FV dose administration and on the muscle state during FVT.

#### 4.5.1. Vibration Frequency and Amplitude

Firstly, by analyzing Table 1, an evident datum is that the most adopted vibration frequency is around 100 Hz, even though several trials adopted stimulus frequencies at 300 Hz [30,33,38,39]. Moreover, only a single study shows after-effects by applying a frequency lower than 100 Hz, however its follow-up is the shortest one (24 h) [34]. Stimulus frequency may be a critical factor, because the relative spindles neural inflow is reported as efficacious to induce changes in spinal alfa-gamma loop [48] and supra-spinal activation [34]. Moreover, spindle afferents are highly responsive to a vibration frequency range of 80–120 Hz, showing a responsiveness of about 1:1 [7,8]. On the other hand, the amplitude is a parameter that largely varyies in the different studies, but less relevant since the stretch-amplitude range, able to elicit a spindle afferent discharge, is extended between a few microns and 2 mm [9]. Moreover, this parameter may be largely altered and decreased by the soft tissue, interposed between the vibrating device and the muscle spindles.

#### 4.5.2. Dose Administration

The single session of FVT lasted from a minimum of 10 consecutive minutes, to 1 h. Moreover, the single sessions were differently distributed in the time. Two main schemes can de individuated. The first one, homogeneous in different researches [30,32,34,36,37,41], consisted of short single sessions, lasting only 10 min, repeated three times/day, separated by only 1–2 resting minutes. In the second scheme, sessions were carried on once a day [31,33,34,35,38,39,40], in this scheme each session lasted 15 [33], 30 [31,35,38,40], or 60 min [34,39]. Finally, the two schemes showed a different number of stimulation days. The first one is concentrated in three sessions/day, during only three consecutive days [30,32,36,37,41]. The second scheme show a more dispersed FVT, being the sessions variously distributed throughout the time, that is, one session/day, during three or five days/week and during 2–26 consecutive weeks [31,33,34,35,38,39,40]. Results seem comparable about the magnitude, whereas the after-effect persistence is, perhaps, favored by the shorter but more concentrated protocol.

#### 4.5.3. Treated Muscles

Despite the variety of the treated muscles in the different studies, they share some common aspects. Firstly, all the treated muscles are anti-gravity and, moreover, inferior limb muscles are clearly privileged (10 studies, out of 12). *Quadriceps* largely constitute the preferential target, both for the number of studies (8) and tested subjects (239) [30,32,33,35,36,37,40,41]. *Triceps surae* [34,35] and *Tibialis anterior* were tested in only two studies, respectively involving 65 and 80 individuals [33,34]. Whereas, in the superior limb, *Biceps brachii*, *Deltoid* and *Pectoralis* were examined in two studies, including 58 participants [31,38]. Two studies applied FVT respectively on two [38] and five [31] muscles in the same session, whereas, all the others reported the treatment on one muscle.

#### 4.5.4. State of the Muscle during the Treatment

A further difference in the various protocols is the muscle condition during FV application: relaxed or contracted. Some studies requested a slight voluntary muscle contraction [30,31,32,33,36,41], in others the muscle is completely relaxed [31,34,35,38,39,40], whereas, in other experimental protocols, either the two conditions were tested and compared [30,32,41]. In this respect, the more persistent effects seem to be elicited in studies where FVT was executed during a voluntary, isometric muscle contraction. The research that explored FV after-effects both in contracted and relaxed condition, with the same protocol [32,36,41], obtained effective significant after-effects only in the contracted state. It might be observed that in relaxed conditions a good mechanical coupling is not performed between the stimulatory device and the stressed anatomical part. Consequently, the mechanical energy applied by vibrator can be altered and dissipated because of a dumping effect. Therefore, the physical characteristics of the applied stimulus may be unknown, not quantifiable and, on some occasions, the magnitude might be not adequate to properly elicit the spindles [49]. On the contrary, a state of muscle contraction, even if at a low level, will increase the segmental rigidity of the vibrated section and will improve the mechanical coupling, ensuring a more faithful propagation of the vibratory energy transmission and in frequency characteristics [50]. It must be noted that FVT after-effects were studied by Transcranial Magnetic Stimulation (TMS) in the motor cortical areas, in healthy and diseased individuals. These studies showed significant rearrangements of the motor cortex, only when FV is performed on a contracted muscle, being absent in a relaxed muscle condition [51,52]. These last studies supported the necessity of the voluntary muscle contraction, during vibration, with references specifically related to the induction of the cortical plasticity. Despite this, now, it is not possible to clarify if, in which conditions and how much the muscle status (relaxed/contracted) influenced the FVT results.

### 4.6. Suggested Physiological Mechanisms

As is well known, mechanical vibration, if directly applied to muscle or tendon, is an adequate way to achieve an intense and highly specific spindle activation and the selected literature is in agreement in considering the muscle spindle activation as an essential step in the development of muscular functionality after-effects [5]. Moreover, neurophysiological studies show that an appropriate FVT can induce a central neural rearrangement able to explain long lasting motor performance changes [51,52,53,54,55,56,57,58].

Now the question concerns how an intense proprioceptive activation could induce such remarkable effects in muscle strength or power, fatigue and balance, thus implying a main central nervous system role in the observed after-effects and whether other mechanisms can be involved.

It should be considered that several papers, adopting FVT, showed statistically significant motor changes a few minutes after treatment ending [35,38,39,40,41]. In these instances, it is unlikely to consider possible morphological changes as being responsible for the observed immediate improvements, whereas it is strongly suggested that a primarily and direct influence of the proprioceptive activation on the central or local motor command, as much as motor units recruitment, firing frequency, synchronization and coordination between agonist and antagonist muscles [59,60]. The persistence (days or weeks) of the after-effects supports the hypothesis that a primary role of the central nervous system influenced, in some way, by the FVT. At the same time, a neural rearrangement, especially at a cortical level, might explain the cross effects also detected on the non-treated limb [30,39]. The longer persistence (months) of motor changes may be due to a spontaneous enhancement of the daily spontaneous activities [32,37], or of a more efficacious training consequent to motor changes induced by FVT [41].

With regard to the motor performance improvements, it is well known that the interplay between agonist and antagonist muscle activities, in a system composed of several muscles, bone segments and joints, take on a determinant role from both the biomechanical and motor task efficiency point of view [61]. Such improved centrally managed joint efficiency might also be a direct reason for the decreased fatigue. Such observations are supported by several studies with TMS which showed a rearrangement of the cortical excitability, selectively involving the motor areas pertinent to the treated muscles after FVT [52,53,54,55,56,57,58,59]. In particular, Marconi’s group, after FVT, applied according to a clustered modality, evidenced a long-lasting advantageous cortical modulation of the agonist-antagonist muscle balance, correlated to the improvement of motor coordination of the joint belonging to the vibrated muscle and its antagonist [51,52]. It is worth noting that motor improvements, induced by traditional physical trainings, are correlated with similar rearrangements of cortical excitability [62,63,64].

From a biomechanical point of view, during movement, the main role of an antagonist is to provide a braking force finalized to control both time and accuracy [65,66,67,68] and, in turn, has a protective role for the joint by modulating its stiffness [69]. Therefore, during the movement the antagonist force output may exceed the level necessary to brake, in particular when the acceleratory phase is ending, and this occurs in especially in non-skilled subjects [70]. Consequently, the agonist-antagonist coordination, by modulating the joint functional stiffness, allows us to manage the movement smoothness [71]. In turn, the smoothness amelioration in a motor task is considered as a direct expression of an improved motor coordination and, consequently, of a more efficacious motor execution [72]. In this regard, Aprile et al., analyzing the movement smoothness, evidenced significant improvements after FVT [38]. This is coherent with Marconi’s neurophysiological data [51,52] and might additionally explain the outcomes of the jumping test where performance is a direct consequence of agonist–antagonist interplay [32,36].

Finally, the ameliorated balance of agonist/antagonist activation might allow a more efficient movement, according to an energetic point of view by inducing a reduction of the internal work in favor of the external one [73]. Such considerations may explain the capacity to perform the exercise with an increase of workout after FVT [37], as well as a lesser fatigability in isotonic movements [30,38], a reduction of the RFD [30] and the improved body stability evidenced by testing the balance in static and dynamic conditions [32,37]. The datum showing a significant decrease of the joint laxity, after FVT, also appears to be coherent and relevant [36].

Nevertheless, it must be underlined that, if the cortical agonist/antagonist is supported by experimental evidence, we cannot exclude other, still unidentified but possible mechanisms, both in subcortical and spinal motor networks and in the sensory function. FVT likely involves spindle afferents and it is known that their influence on motoneuron behavior, that is, both by favoring the recruitment [9] and by modulating the strength output decline [74]. Muscle strength enhancements, in sportive training, are improved by changes in alpha-motoneuron discharge [60,61]. Finally, the space perception results should be modified after FVT, this being potentially relevant for the balance improvements [75,76].

## 5. Conclusions

This review shows that FV seems to be a brief and efficient stimulus able to enhance the conditional capacities in healthyindividuals. These changes could be attributed to a better agonist/antagonist interplay because of a rearrangement in central and segmental nervous pathways. Consequently, power and absolute force is increased, kinematic is smoother and articular efficiency is higher, whereas force output declines as time is delayed. All listed papers showed long-lasting positive motor changes, without negative collateral effects, the FVT acting as a motor conditioning. Table 1 shows that FV can be administered with several treatment protocols and properties varying mechanical stimulus characteristics, stimulus administration modalities and intervention duration as evident from the paper selected. Between them, based on effects duration, administration modalities and motor improvements, the more efficient FVT seems to be characterized by a stimulus frequency of about 100 Hz, a three–five days administration, on a contracted muscle.

## Figures and Tables

**Figure 1 jfmk-06-00039-f001:**
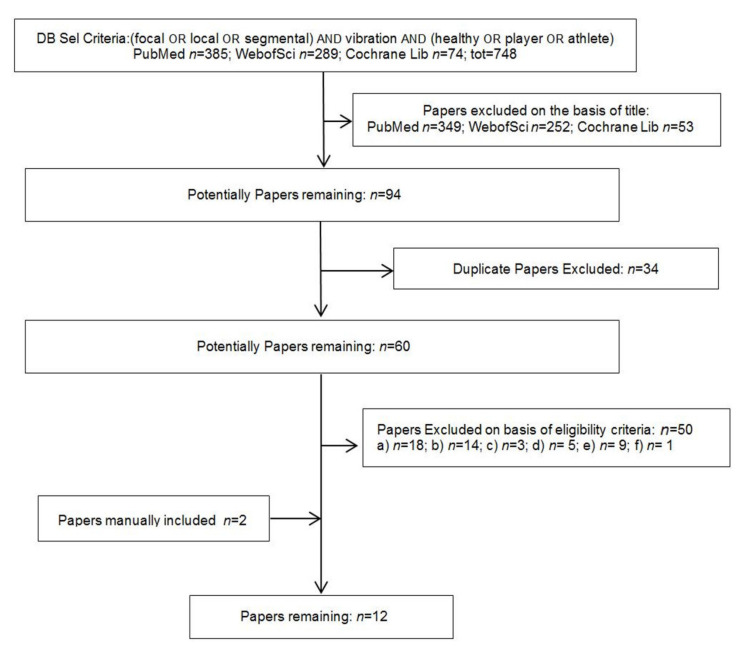
Flow diagram of studies selection process.

**Table 1 jfmk-06-00039-t001:** Main characteristics of the studies included in the review. The outcome values at the end of the follow-up are reported (* *p* < 0.01; ** *p* < 0.001). ^1^ Fatigue index: fatigue evaluation computed as muscle fiber velocity conduction. ^2^ Postural Sway Area: the area which encloses the data points relative to oscillations in the horizontal plane of the center of gravity even when a person is standing. ^3^ Velocity of Sway: velocity of the center of gravity oscillations.

Study	Tested Sbjts	Age (Years)	Main Protocol Features: FV Frequency; Application Duration; Num appl; Mechanical Signal Amplitude	Body Part Treated & Muscle Status during Treatment	Follow-up	Outcomes at the Follow-Up End (POST vs. PRE)
Fattorini L. et al., 2006 [30]	21 (6 F; 15 M)	31.0 ± 2.4	100 Hz, 3 applications/day, lasting 10 min each one, repeated for 3 consecutive days; p-p ≈ 0.05–0.15 mm	Quadriceps (one leg); Contracted/relaxed	2 weeks	RFD ≈+ 27% *; Fatigue resistance (isotonic conditions) ≈+40% ** (vibrated side) ≈+15% (untreated side) MVC Unchanged; No changes with muscle relaxed during treatment
Casale et al., 2009 [31]	10 (M)	25–50	300 Hz, 1 application/day, lasting 30 min, repeated for 5 consecutive days; p-p ≈ 2 mm	Biceps brachii (one arm); Relaxed	48 h	Fatigue index^1^ ≈ −20% * MVC Unchanged
Filippi GM et al., 2009 [32]	60 (F)	60–72	100 Hz, 3 applications/day, lasting 10 min each one, repeated for 3 consecutive days; p-p ≈ 0.2–0.5 mm	Quadriceps; Contracted/relaxed	90 days	Power ≈ +35% * Squat Jump height ≈ +55% * Postural Sway Area^2^ ≈ –35% * Velocity of Sway^3^ ≈ –35% *; No changes with muscle relaxed during treatment
Pietrangelo T et al., 2009 [33]	9 (5F; 4M)	71.0 ± 5.7 (F); 75.3 ± 6.9 (M)	300 Hz, for 15 min, 1 application/week for 8 weeks and 3 applications/weeks during the subsequent 4 weeks; Amplitude unknown	Quadriceps; Contracted	16 weeks	MVC ≈ +40% * (M) ≈ +50% * (F); Muscle fiber cross-sectional Area: Unchanged Fast Myosin isoforms: +12% *
Lapole et al. 2010 [34]	29 (Gender unknown)	21.7 ± 1.7 College students	50 Hz, 1 application/day, lasting 1 h, repeated for 14 consecutive days; p-p ≈ 0.2 mm	Triceps surae; Relaxed	24 h	MVC ≈ +6.9% * EMG ≈ +9.3% * Twitch properties: Unchanged Muscle mass: Unchanged
Iodice et al., 2011 [35]	36 (Gender unknown)	21.5 College students	300 Hz, 1 application/day, lasting 30 min, repeated 3 time/week, for 4 consecutive weeks: p-p ≈ 2 mm	Quadriceps Gluteus maximus Biceps femoris, GastrocnemiusTibialis anterior; Relaxed	2 months	MVC ≈ +32.5% * Torque ≈ +33% *
Brunetti et al., 2012 [36]	18 (F)	22,7 Volleyball players	100 Hz; 3 applications/day, lasting 10 min each one, repeated for 3 consecutive days: p-p ≈ 0.3–0.5 mm	Quadriceps; Contracted/relaxed	240 days	Squat Jump height ≈ +26% ** Counter Movement Jump height = +13% **Knee Laxity ≈ −18% **; No changes with muscle relaxed during treatment
Brunetti O et al., 2015 [37]	40 (F)	65.2 ± 3.0 Postmenopausal women	100 Hz, 3 applications/day, lasting 10 min each one, repeated for 3 consecutive days; p-p ≈ 0.3–0.5 mm	Quadriceps; Contracted	360 days	Power ≈ +40% ** Squat Jump height ≈ +40% ** Postural Sway Area^2^ ≈ -35%, **
Aprile et al., 2016 [38]	48 (36F; 12M)	25–50	100 Hz, 1 application/day, lasting 30 min, for 3 consecutive days; 200 Hz, 1 application/day, lasting 30 min, for 3 consecutive days; Pressure 150 mBar	Deltoids, Biceps Pectoralis; Muscle status unknown	10 days	Fatigue resistance: ≈ +30%* (@200 Hz) Unchanged (@100 Hz) Movement Smoothness: ≈ +27% * (@200 Hz) Unchanged (@100 Hz)
Souron et al., 2017 [39]	44 (24 F; 20 M)	20 ± 1.0	100 Hz, 1 application/day, lasting1 h, repeated for 3 days/week, for 8 consecutive weeks; p-p ≈ 1 mm	Tibialis anterior (one leg); Relaxed	2 weeks	MVC ≈ +12% ** (vibrated side) MVC ≈ +10% * (untreated side)
Feltroni L, et al., 2018 [40]	27 (12 F; 15 M)	22.2 ± 2.7	80/300 Hz, 1 application/day, lasting 30 min, repeated for 5 consecutive days. Pressure 240 mBar	Quadriceps; Relaxed	4 weeks	Peak torque ≈ +29% * (both at 80 and 300 Hz)
Filippi GM et al., 2020 [41]	28 (M)	24 ± 3.0 University & phd students	100 Hz, 3 applications/day, lasting 10 min each one, repeated for 3 consecutive days. p-p ≈ 0.2–0.5 mm	Quadriceps; Contracted	2 weeks	Peak Power ≈ +10% * Average Peak ≈ +7% * Total Work ≈ +8%*

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
