# Peer review of "Is the Focal Muscle Vibration an Effective Motor Conditioning Intervention? A Systematic Review"

_jfmk, 2021, doi:10.3390/jfmk6020039_

Round 1
Reviewer 1 Report
INTRODUCTION
Lines 33-35 : in addition to problem of compliance, some subject with muscle frailty or exercise intolerance (e.g. elderly, clinical population) could not fit with classical resistance training design and could then benefit from vibration. Maybe this could be detailed in the manuscript
Line 36: is there any study showing that training compliance is lower with “classical” training program than with WBV training program?
Line 40: “that” rather than “as”
Line 43: define what is CNS
Maybe be more precise on the central parts that are activated by vibration stimulus
“Due to the fact that the neu-44 romuscular spindles exert a decisive influence on the control of motor activity and a subsequent central adaptation [10] and this inducing changes in motor functionality [11].” This sentence has no sense, please reword.
Line 49: you need to add references here!
Line 50: what are the advantages of focal vibration compared to WBV?
Line 54: same here, you need to add references
Line 55-69: there is no ref (!!!!!) in this whole paragraph. This article is a review, then you need to add the references of the paper you’re referring to.
“Due to the possible FV applications, in the last period there have been realised sev-70 eral reviews on the FVT”. This sentence has no sense, please reword
Lines 71-72: what is “enduring motor effects” ?
<line 86: “healthy” not “health”
METHODS
Paragraph 2.4: you need to give more details here. What were the items? How can you judge the quality considering these items ?
Ligne 164: why are you reporting only significant results? Is not the aim of a review to present every results, including the non-significant ones?
There are studies missing in the Table 1. For instance this ref: Souron R, Besson T, Lapole T, Millet GY. Neural adaptations in quadriceps muscle after 4 weeks of local vibration training in young versus older subjects. Appl Physiol Nutr Metab. 2018 May;43(5):427-36.
You can check this narrative review on local vibration to find potential other articles to include: Souron R, Besson T, Millet GY, Lapole T. Acute and chronic neuromuscular adaptations to local vibration training. Eur J Appl Physiol. 2017 Aug 01
Please delete the “?” that we can find sometimes in the table 1. Or explain why you put this symbol.
Lines 172-179: I am not very sure that this paragraph has its place here. Maybe more in the discussion. Further, what do you mean by “describre the interface apparatus body”? I do not see any interest in describing that. The more important information when FV is applied is 1) the frequency, 2) the amplitude, 3) the duration of application and 4) the muscle where FV is applied. In that way, I guess that most paper give these information.
Lines 199-200: are you sure about the %??? It seems quite high. And we need more info on the way you calculated this %
Paragraph 4.2: why are you talking about fatigue? You stipulate that the aim of this review is to only focus on the long-term effect of FV on physical performance. Which is not the case with fatigue that is more a response in the short term.
Lines 241: I am really really surprised by those results. As you said previously in the manuscript, did these authors controlled for confounding factors that could have influenced this results, e.g. changes in the volume of physical training. I cannot see how a 3-days application could lead in improvement up t 240-360 days.
Line 355: what do you mean by “metabolic and morpholigal changes”? Do you really think that a prolonged FV application could lead in morphological changes? If yes, please give some food for that
This paper lacks of a “perspectives” part.

Reviewer 2 Report
March 15, 2021
Manuscript ID: jfmk-1152182
Type of manuscript: Review
Title: Is the Focal Muscle Vibration an Effective Motor Conditioning Intervention? A Systematic Review
Authors: Luigi Fattorini *, Angelo Rodio, Vito E. Pettorossi, Guido M. Filippi
Submitted to section: Kinesiology and Biomechanics,
https://www.mdpi.com/journal/jfmk/sections/Kinesiology_Biomechanics
Title: Is the Focal Muscle Vibration an Effective Motor Conditioning Intervention? A Systematic Review
Brief Summary of the Manuscript:
The manuscript reviews results of studies in which focal vibration was administered to “normal” human participants”. The authors selected twelve manuscripts from a total of 748 meeting select criteria during a systematic literature search of three scientific databases and other methods. Two authors extracted relevant data according to a structured rubric they applied to each study. Inclusion criteria were selected by (a) population: healthy subjects; (b)intervention: treatment adopting localized mechanical vibration with a clear description of apparatus; (c) outcomes:neuromuscular parameters regarding condition abilities; (d) effects duration: at least 24 hours from treatment ending. Study criteria were summarized in a table. The authors then compared and contrasted findings reported among similar studies. They summarized effects on maximal muscular strength, power, and fatigue. However, as the authors not in “Bias”, the studies varied greatly in participants condition, age, focal vibration (FV) protocol used, time between FV application and testing, and outcomes measured. The authors end the discussion with suggested physiological mechanisms.
Keywords: Power; Force; Efficiency; Motor Adjustments; Muscle Vibration
Broad comments:
The topic is of great interest. The title captures the reader’s interest. The title uses “Effective Motor Conditioning Intervention”. In the text the reader would like to know exactly what that term means. Unfortunately, the following words are used without the reader understanding the specificity, difference, or sameness for each term. The readers think they know where the manuscript is going but there are so many terms for the same idea (motor system, motor functionality, motor activity by an analytic functional intervention, motor performance, and motor effects). It would be good if you could draw the connection between FVT and Effective Motor Conditioning Intervention through the manuscript.
“Purpose: This paper is aimed to review these studies and to characterize the FVT effectiveness on long-term conditional capacities in relation with FVT characteristics. A systematic search of studies published from 1985 to 2020 in English. “
“Purpose: on efficacy of sole FVT to elicit long-term motor performance improvements in healthy subjects. Moreover, this literature review tries to clarify the dose-effect relationship between the motor improvements and the intervention characteristics, as stimulus frequency, administration modalities and muscle state during treatment.”
The manuscript will benefit greatly if someone familiar with English edits the grammar and sentence structure. Sentences often need verbs. The editor can fix the incomplete sentences, sentence fragments, and run-on sentences.
Make sure all abbreviations are spelled out the first time they appear in the text.
Abbreviations used:
focal vibration (FV)
focal vibratory (FVT)
CNS
whole body (WBV)
maximal voluntary contraction (MVC)
rate of the force development (RFD)
counter movement jump (CMJ)
Wingate test (WnT)
Undefined terms:
Fatigue Index
SJ height
Sway area
Introduction. The information needs structure. It states types of vibration and exercise training in general terms; we want you to tell us about FV. Be specific. “Focal Vibration” is a concept or a technique which needs explaining. What type of equipment is traditionally used? Does it hurt? Why should we be excited about it? Many purposes are stated in the Introduction. The reader wants to understand the main purpose of this review. Why are you writing this? What will I learn if I read this manuscript? Tell us the purpose and then structure the rest of the manuscript around that one purpose.
- Materials and Methods.
Line 129 2.4. Assessment of methodological quality – Tell us more about how you got from the checklist to a qualitative score.
- Results
Many reviews include a flow diagram as you have presented in Fig 1. Because you clearly explain this your selection criteria, Fig 1 is unnecessary but it is up to you and the editor.
Line 150. Downs and Black modified scale needs to have the reference cited.
- Discussion
I have attached a paper to assist you. Murillo N 2014. You will notice it is similar in structure to your manuscript but much of what you have in the Discussion, Murillo has placed in the Results section. These paragraphs address your purpose.
Specific comments
Line 25. “The after-effects persistence ranged between 24 hours and several months.”
Because you preselected papers with “outcomes time analysis enduring more than 24 hours”.
Line 27. “Short FVT protocols enduring few days seem to be more effective to induce persistent improvements enduring weeks or more.” This is a strong statement to put at the end of your abstract given you say: Line 416. “All listed papers showed long-lasting positive motor changes, without negative collateral effects.”
Line 119. 2.3. Study Eligibility
The authors describe five criteria for excluding studies from their review.
“Studies were excluded if they (a) analyzed outcomes in a short follow up period (<24h);” GOOD but make this time limit clean in your
Line 129 2.4. Assessment of methodological quality
The authors then explain how they evaluate whether a manuscript is worthy of being included in their review. They apply criteria contained in a 16-item checklist. Quality scores were then assigned. The reader would like to know more about the checklist and how scores were assigned and then converted to qualitative scores.
Line 144 Fig 1. is not necessary. This information is contained in the text.
Line 166 Table 1. Main characteristics of the studies included in the review.
Please provide a legend for Table 1. It should include an explanation of all abbreviations.
Also, the reviewer is not able to understand whether the results were positive and persistent. Results listed in the column, “Main Significant Results at the Follow-Up End” of Table 1., list only one time point. Sometimes results indicate “RFD = +25%”. This sounds positive. Only result is listed as: “Fatigue index = -20%.” That sounds as though it could be better or worse than a control.
Line 170. 3.3. Bias (Often named Limitations)
The reader agrees with the authors. This should go in the Discussion Section.
- Discussion. “The aim of this review is to focus the attention on persistent motor effects after a sole FVT appliedon healthy individuals and to describe the relationship between protocol and performance changes, by analyzing the intervention modalities.”
Line 195 4.1. Outcomes
The reviewer noticed that of the 12 studies reviewed, seven applied FVT repeatedly. FVT was not applied only once as is expected for a “sole FVT” treatment. Sole means “only one” or “bottom of the foot”
Line 182. It is difficult to know whether all participants were healthy. How would you describe “healthy”? Uninjured? Are trained athletes the same as those who do not participate in athletics or sports?
“In all of these, stimulus characteristics and administration protocols were well defined, and the majority (10 outof 12 studies) compared the treated individuals with an untreated control group.”
Data consist of a table which lists studies individually and summaries of what similar results were presented among the studies. The authors know the details of the studies. In the Table, it reads as though most studies compared one limb which received FVT while the opposite limb did not. These are presented as the experimental limb and the Control limb. The treated and untreated “individuals” come in the later summaries. Would you clarify the Table data in the Table legend?
“It seems primarily and immediately relevant that all trials reported positive and persistent neuromuscularchanges and no one of the tested individuals reported negative collateral effects.”
- Conclusions
Line 416. “All listed papers showed long-lasting positive motor changes, without negative collateral effects.”
Line 119. Are you sure your inclusion did not preselect for this conclusion? “criteria were selected by (a) population: healthy subjects; (b) intervention: treatment adopting localized mechanical vibration with a clear a description of apparatus; (c) outcomes: neuromuscular parameters regarding conditional abilities; (d) effects duration: at least 24 hours from treatment ending.”
Line 190. By selecting “healthy subjects”, did you possibly eliminate papers with negative effects?

Reviewer 3 Report
This paper titled “Is the Focal Muscle Vibration an Effective Motor Conditioning Intervention? A Systematic Review” is an informative systematic review. The authors reviewed the studies documenting the effects Focal Vibration (FV) therapy. It is interesting that the authors focused on FV therapy only in the context of healthy individuals. This provides a potentially very useful insight in an area where most reviews focus on FV therapy for individuals with motor impairments. This review focuses on healthy individuals and only those that received FV only, and no other intervention. This helps understand the effects of FV in isolation.
Overall, this systematic review has been performed well. The authors have provided insightful details and discussions on the papers reviewed. In general, the methodology has been thorough. There are small but important details that are missing. Other than that there are some minor corrections. All of those are listed below.
- Table 1: In my opinion, three things define FV therapy: 1) frequency, 2) measure of intensity (amplitude or acceleration) 3) duration and schedule (may be called protocol). Please add intensity information in this paper as it is missing. This can be amplitude, acceleration, or whatever else is available for each reference. Almost every FV paper in the literature will quote this information. If this information is not available for a specific paper, please mention that (e.g., “amplitude unknown”). Ideally, intensity information should be added under the “main protocol features” column in Table 1.
- Please mention somewhere in the paper why the FV intensity was not considered for this review as this is generally considered an integral part of FV. One or two sentences are sufficient.
- Not all of the studies in this paper have entirely randomly selected healthy populations. The authors should provide additional information about the population, if there is the possibility of bias towards a particular group. It should be brief. For example, for [14], this information would be “elderly with sarcopenia,” for [16], “volleyball players,” for [17], “college students,” for [19], “postmenopausal osteoporotic,” and so on. This could be useful in comparing and contrasting the results for different studies in this review paper. This could be under Section 3.3 (bias) or elsewhere in the text, or in Table 1 under “subjects,” or in a separate table.
- This is only a recommendation, but it would be nice if the authors provide names of the devices used in different studies. For example, [18] used a device called the “Cro System.” This could provide some potentially useful insights to the readers. This information can be provided in Table 1 (under main protocol features) or it could be provided in a subsection of text, or it could be in a different table. If the device name is not provided by a source, then just list “device unknown” for that source.
- Table 1: For references [21] and [23] the muscle status is listed as “Relaxed?” which is confusing. Is this a typo? If yes, then please remove the question mark. If this is not a typo, and the question mark indicates that the information is not clear from the paper, then instead of “Relaxed?” write something like “muscle status unknown.” You should be specific and clear.
- Fig. 1: Please remove the red error underline from under “sel” and “WebofSci”
- Line 13: Entire abstract. Please check grammar and sentence construction. Try to have shorter, clearer sentences where suitable.
- Line 25: “Short FVT protocols enduring few days seem to be more effective…” than what? More effective than short protocols lasting a few minutes? More effective than long protocols lasting a few weeks? Or are you trying to say protocols that last at least a few days are adequate for inducing persistent improvements endearing weeks or more? Please clarify.
- Line 51: Please correct the phrase “single o few muscles”
- Line 70: Entire paragraph, several issues with grammar and sentence construction, please make corrections
- Line 86: Please correct the phrase “only health individuals”
- Line 201: Please correct “did not increased” to “did not increase”
- Line 257: Please correct “note of worthy” to “noteworthy”
- Line 263: Please correct the grammar in this sentence.
Round 2
Reviewer 1 Report
The article has been well improved and I have no further comment.
My only suggestion is about the English level. I think that the article will gain in clarity if a native speaker doubled checkt it.
Reviewer 2 Report
The authors have addressed all concerns I had with preparation of the manuscript. It is now in good form. The meaningful data is clearly described, presented accurately, and summarized with justified interpretations. I approve publication the current document.